# Proposing a Framework for the Restorative Effects of Nature through Conditioning: Conditioned Restoration Theory

**DOI:** 10.3390/ijerph17186792

**Published:** 2020-09-17

**Authors:** Lars Even Egner, Stefan Sütterlin, Giovanna Calogiuri

**Affiliations:** 1Citizens, Environment and Safety, Institute of Psychology, Norwegian University of Science and Technology, 7048 Trondheim, Norway; 2Faculty of Health and Welfare Sciences, Østfold University College, 1757 Halden, Norway; stefan.sutterlin@hiof.no; 3Division of Clinical Neuroscience, Oslo University Hospital, 0450 Oslo, Norway; 4Faculty of Health and Social Sciences, University of South-Eastern Norway, 3045 Drammen, Norway; giovanna.calogiuri@usn.no; 5Department of Public Health and Sport Sciences, Inland Norway University of Applied Sciences, 2411 Elverum, Norway

**Keywords:** restorative environments, conditioning, attention restoration theory, stress reduction theory, perceptual fluency account, nature-based recreation, nature exposure

## Abstract

Natural environments have been shown to trigger psychological and physiological restoration in humans. A new framework regarding natural environments restorative properties is proposed. Conditioned restoration theory builds on a classical conditioning paradigm, postulating the occurrence of four stages: (i) unconditioned restoration, unconditioned positive affective responses reliably occur in a given environment (such as in a natural setting); (ii) restorative conditioning, the positive affective responses become conditioned to the environment; (iii) conditioned restoration, subsequent exposure to the environment, in the absence of the unconditioned stimulus, retrieves the same positive affective responses; and (iv) stimulus generalization, subsequent exposure to associated environmental cues retrieves the same positive affective responses. The process, hypothetically not unique to natural environments, involve the well-documented phenomenon of conditioning, retrieval, and association and relies on evaluative conditioning, classical conditioning, core affect, and conscious expectancy. Empirical findings showing that restoration can occur in non-natural environments and through various sensory stimuli, as well as findings demonstrating that previous negative experience with nature can subsequently lower restorative effects, are also presented in support of the theory. In integration with other existing theories, the theory should prove to be a valuable framework for future research.

## 1. Introduction

The effects of the environment on human perception, cognition, affect, and behavior have been the subject of psychological research for several decades. Especially natural environments have been investigated for their restorative potential [1]. Restorative mostly translates into the psychological and physiological advantages of being exposed to an environment, often nature [2]. This includes walking in [3], having a window view of [4], or even just viewing pictures of nature [5]. The restorative environments field is currently dominated by two leading theories of how nature exerts these positive effects: the attention restoration theory and the stress-reduction theory.

Attention restoration theory (ART) [5,6] has a primary focus on attentional processes and cognitive fatigue. More specifically, ART focuses on the process of depletion and restoration of *directed attention*, a form of attention on which people rely when they focus on tasks that are not spontaneously interesting (e.g., writing a work-report or driving in the traffic). Directed attention is a *limited* cognitive resource and its depletion typically leads to cognitive fatigue. Restoration of directed attention occurs by resorting on a spontaneous (and, thus, effortless) form of attention, which in ART is referred to as fascination. Some environments are particularly effective in triggering fascination and, thus, restorative processes, which are associated with positive affective responses. In particular, because of their typical characteristics (e.g., clouds moving in the sky or leaves swooshing in a breeze), natural environments can trigger spontaneous attention avoiding excessive emotional arousal, or so-called soft-fascination. Alongside fascination, ART propose that three other environmental qualities can elicit attention restoration processes: being away (the extent to which an environment provides the opportunity to get away from daily hassles), compatibility (the extent to which an environment matches a person’s inclinations at a given point), and extent or coherence (the extent to which the elements of an environment are connected in an orderly fashion). It should be noted, however, that ART does not associate attention restoration processes exclusively with natural environments; other environments can trigger fascination as well, as long as they provide certain characteristics. Studies on ART often operationalize this attention restoration process (and thus assess the restorative effects of nature exposure) predominantly through assessments of perceived environmental qualities and psychometric tests on attention. For instance, Hartig [7] developed an ART-based questionnaire to assess the restorative potential (or perceived restorativeness) of different environments. Studies have also shown that exposure to nature can elicit attention restoration processes as compared with exposure to urban environments [3,5].

Stress reduction theory (SRT) [8], another commonly used theory in the field of environmental restorativeness, focuses on psychophysiological stress. Not unlike ART, SRT is grounded on a psycho-evolutionary perspective, i.e., it assumes that human beings are (still) innately bound with the natural environment as their habitat, while being not fully adapted to modernized urban settings, which (on an evolutionary perspective) appeared recently in human history. This assumption is in line with Wilson’s biophilia hypothesis [9], according to which humans, as a product of evolution, have an innate affinity with living things and the natural world in general. In SRT, this implies that, on the one hand, environments devoid of nature are perceived as sources of potential threats and leading to a stress response, and, on the other hand, elements of nature such as water and trees elicit positive affective responses, an evolutionary adaptation that would increase the chances of survival. The combination of these processes results in a more rapid reduction of psycho-physiological stress when people are exposed to scenes of nature, as compared with no exposure to nature. For example, in two classic studies, Ulrich showed that patients had on average 9.3% quicker recovery from surgery if they were placed in rooms with a window showing a view on nature vs. windows showing a featureless brick wall [10], and that individuals recovered more quickly from a stressful situation if they viewed a video showing natural landscapes vs. a video showing urban landscapes [11]. Although SRT shares with ART a psycho-evolutionary foundation, studies based on SRT differ substantially in their way of framing stress and attentional processes. In fact, they most commonly operationalize the restorative effects of nature exposure through physiological indicators of stress (e.g., heart rate and skin conductance). Moreover, unlike ART, SRT assumes that natural environments do not require attention to elicit stress recovery; in fact, recovery would best occur in presence of natural environments that trigger mild-to-moderate interest, even when these are in the background (e.g., as a view from a window).

Stress reduction, or relaxation, has been shown to activate the parasympathetic nervous system, which engages bodily “rest-and-digest” functions [12]. The parasympathetic nervous system facilitates vegetative, non-emergency responses. It affects bodily functions such as decreased heart rate, increased digestive activity and generally conserves activity [12]. If the perception of nature does induce relaxation, this would increase parasympathetic nervous system activity, which in turn could facilitate physical recovery. The most specific theory underlying parasympathetic nervous system activity facilitating physical recovery is Porges [13] and the concept of neuroception, the unconscious perception of safety (see Section 2.7 for a more detailed discussion on this).

Alongside these theories, alternative underlying mechanisms explaining the positive affective responses associated with nature exposure have been proposed. Among the most commonly cited, the role of perceptual fluency, which introduced the concept of fractals, has been emphasized [14]. Fractals are described as self-similar patterns emerging at different scales, which are typically abundant in natural settings as opposed to urban or other built environments. For example, all tree branches are scaled-down versions of the tree itself. Perceptual fluency suggests that the natural environment is processed more fluently than urban environments, and this difference would explain the greater restorative potential of the former.

### Challenges and Limitations of Current Theories

Although the theories described above are largely used in the field of environmental psychology to interpret restorative processes, some have raised concerns about specific aspects of these theories [15]. As mentioned above, ART and SRT are grounded on a psycho-evolutionary perspective. Evolutionary psychology is often employed to explain the existence of traits, but we argue that explaining phenomena with ancient theoretical situations is an unfavorable use of evolutionary psychology. A limitation of such a perspective is that it could be said to not fully capture the nuance of the human–nature relationship. For example, it has been shown that water often has restorative properties [16]. This could be viewed as an evolutionary adaptation: proximity to water is advantageous as it is essential to survival. However, the opposite could also be true: people should be on high alert near water because water is a natural destination for predators. The psycho-evolutionary perspective underlying SRT could be said to be further challenged by studies suggesting that non-natural environments can have restorative effects. For example, restorative effects have been demonstrated in museums [17], hospital rooms decorated to resemble local home environments [18], and other non-natural environments [19]. While ART does indeed contemplate the possibility that non-natural environments can have restorative qualities, we agree with a further point made by Joye and van den Berg [14], who wrote “A detailed analysis of SRT’s psycho-evolutionary framework shows that neither current empirical evidence nor conceptual arguments provide any strong support for the hypothesis of restorative responses to nature as an ancient evolved adaptive trait.” (p. 1). It is important to note that the reliance on evolutionary innate arguments does not weaken the theory itself, but rather does not strengthen it and should not be used as an argument regarding its validity.

Theoretical frameworks not based on evolutionary perspectives such as perceptual fluency, are not subjected to this specific critique, but may still suffer from difficulties in demonstrating a causal relationship between fractals and restorative effects. Nature is considered to be restorative, and it is also characterized by fractal patterns. Therefore, the correlation between nature and fractals might be due to an artefactual relationship, rather than causal effects. Even if no direct *causal* connections exist between fractal patterns and restoration, studies would still show a correlation between fractal patterns and restoration, as shown in studies used to support fractal theory [20].

Another critique moved to ART is that, while it explains the increase in attentional test scores, which is used to operationalize attention restoration, attention does not directly explain some of the broader psycho-physiological effects of nature exposure, such as quicker hospital recovery time [10]; reduced blood pressure, anger, and aggressiveness; reduced arousal measured by cardiac interbeat interval [21]; and lower sympathetic activation measured through skin conductance level [22]. It has been argued that attention restoration could spark stress reduction, which again lower blood pressure and similar [23], but we argue the opposite is better documented. Changes in attention explain physiological effects indirectly through stress, but stress reduction processes may explain both attention restoration and physical recovery directly. For example, a short five-day meditation program focusing on relaxation increased scores on executive control [24], a psychometric often used to suggest attention restoration. ART does posits that exposure to nature elicits meditative states, and studies have shown that people tend to spend more time in meditative states when they are in nature as compared with when they are in urban environments (see, e.g., the study using a mobile electroencephalogram [25]), which can provide a link between the cognitive benefits of the physiological responses to nature exposure. However, questions have been raised regarding the validity of comparing natural environments with urban environments. For example, in their systematic review, Bowler et al. [26] stated: “Differences between a natural environment and an alternative environment could arguably be due to factors of the alternative environment rather than those of the natural environment. For instance, an outdoor built environment might provide additional stresses, such as traffic, which do not feature in a natural environment” (p. 8). In other words, the reduction of physiological stress observed in nature may not be explained by meditative states triggered by nature exposure (or other direct restorative effects of nature), as much as to the lack of stressor. Corroborating this assumption, unlike reviews and meta-analysis that have included comparisons of natural vs. urban environments [26], those that have specifically compared indoor vs. nature environments have failed to show consistent evidence of psychological and physiological benefits of nature exposure [27,28]. This suggests that stress reduction and attention restoration can co-occur and are not mutually exclusive, but rather associated, thus making the distinction between attentional and stress-related effects of nature exposure postulated by ART and SRT somewhat blurred.

Another limitation of all the theories described above is their heavy reliance on vision, leaving an gap with regard to the role of other environmental factors (e.g., noises, smells/scents, temperature, etc.) in contributing to restorative processes. For example, restorative effects have been found using only nature sounds [22] and, more in general, research on soundscapes has demonstrated the impact of auditory inputs on psychological and physiological indicators [29,30,31]. Scents and other substances in the air of natural environments are also likely to play an important role in restorative processes associated with experiences in nature. For instance, studies have demonstrated the effect of phytoncides, essential oils emitted by plants also referred to as “aroma of the forest”, on stress hormones and the immune function [32,33,34]. The reliance on vision would imply that visually impaired individuals are unable to experience restoration in nature, and no evidence exists to suggest this.

A strong takeaway is that there is more to learn about these specific theories as well as human and nature relationships. The shortcomings of ART and SRT open the door that there could be alternative explanations for the effects seen on human psychology and physiology regarding human and nature relationships. In light of the limitations of the major theories currently in use, the purpose of this paper is to present the conditioned restoration theory (CRT) as a new theoretical framework that may contribute to explaining the mechanisms underlying the restorative effects of nature, as well as other environments. Originating from a discussion in a lecture on environmental psychology, which later was outlined in a master’s thesis [35], the theory was substantial reworked and developed drawing on the different author’s backgrounds from health and environmental psychology. Implementing research from the former to explain phenomena in the latter, while acknowledging that natural environments often are characterized by features that make them particularly advantageous for eliciting psychophysiological restoration, CRT postulates that restoration can also occur as a result of a classical conditioning process. In the following sections, the paper describes CRT, discuss its strengths and limitations, reviews research literature that supports it, and proposes how it can be integrated with established theories.

## 2. Conditioned Restoration Theory

Depending on specific circumstances and settings, CRT proposes that much of the effect of natural environments and other restorative stimuli can be accounted for in a two-step model relying on conditioning and associative learning. The first step involves associating nature with relaxation, and the second step involves subsequently retrieving the same relaxation when presented with an associated stimulus, as shown in Figure 1. First, people are conditioned to associate nature with something relaxing. Then, when subsequently presented with nature, conditioning triggers relaxation.

By “relaxation” and “stress”, CRT refers to their meaning as defined by core affect, considered to reflect the simplest rawest dimensions of emotional feelings [36]. Core affect consists of two dimensions, arousal and valence. Arousal represents activation–deactivation, while valence represents pleasure–displeasure. With “stress” the article is referring to the core-affect state of high arousal and low valence, and with “relaxation” the core affect state of low arousal and high valence.

The theory postulates the occurrence of four stages: (i) unconditioned restoration, an unconditioned relaxation reliably occur in a given environment (such as in a natural setting); (ii) restorative conditioning, the relaxation becomes conditioned to the environment; (iii) conditioned restoration, subsequent exposure to the environment, in the absence of the unconditioned stimulus, retrieves the same relaxation; and (iv) stimulus generalization, subsequent exposure to associated environmental cues retrieves the same relaxation. A similar version of the theory, conditioned place preference, is the foundation of large parts of drug-addiction research [37], and the authors argue that there is strong support for each of these processes, which combined could contribute to explaining most research on restorative environments. In the following sections, each of the stages presented above are described and discussed in light of existing literature.

### 2.1. Unconditioned Restoration

We suggest that the primary reason humans relax in nature is based on the setting in which the majority of members of modern society interact with nature; that is, in relaxing leisure activities. Forests, parks, beaches, and mountains are primarily used for activities such as hiking, family trips, exercise, picnics, and general recreation. While the frequency of interaction with nature is declining, an “extinction of experience”, the current level of visitations to nature could still be said to be high [38,39], although we suspect strong cultural variations exist. These are activities that are relaxing in themselves. Only a fraction of the population associates nature with stressful activities, including but not limited to lumberjacks, fishermen, victims of natural disasters, and soldiers with combat experience in natural environments. Moreover, some of the literature tends to refer collectively to different types of environment as “nature”, but it could be argued that forests, parks, mountains, and lakes are widely different environments. A gardener may only have a work relationship with parks, not with nature as a whole, and thus enjoy relaxing activities in most natural environments. It is important to note that a broad range of “work relations” exist. An outdoor guide’s work relation to forests is most likely more relaxed than that of a logger, which is again more relaxed than that of a seasoned forest fire-fighter. The share of people exposed to natural disasters is increasing [40], but being exposed to floods could only reduce restoration of water, not mountains. Nature is often an environment removed from daily chores, where little “needs to be done”. This is naturally not the case for many indigenous, native, or nomadic people. Some factors can be gender-specific, such as women having a higher fear of crime in urban green spaces [41]. A CRT framework suggests that children who grew up in nature do not experience the same restoration in nature as other children. As relaxation can occur in other places than in nature, CRT claims that several other environments, for instance living rooms, can become as relaxing as nature. This is discussed further in Section 2.2, Section 2.3, Section 2.4.

While it is methodologically difficult to separate leisure activity from leisure environment, and thus say that it is the activity itself that is relaxing, we argue that because leisure has been shown to be relaxing in several environments [17], as well as being generally accepted in the leisure literature to be relaxing [42,43], it is a reasonable assumption to make that leisure activities are in themselves relaxing, and not just the environment they are conducted in. For example, a social scientist counting park visitors for a research project should experience less restoration than when doing leisure activity in that same park.

The second reason posited for why humans relax in nature relies on ART’s person–environment factor of being away [6]. The lack of social feedback in nature has previously been pointed out as key [44]. Although social feedback can be positive or negative, it mostly elicits arousal, while a lack of social feedback could be said to generally lower arousal.

Finally, socially induced expectancy is believed to affect the emergence of relaxation in nature. This argument naturally suffers from causation issues, as social expectancy would unlikely occur if nature were not a relaxing environment in the first place. Nonetheless, it is argued that this strengthens the effect, as expecting something to have an effect has been shown to affect the activity itself [45]. It should be noted that non-social expectancy and conditioning effects are not included here, as personal experience is required for this to occur, and this step is about the emergence of relaxation, not later recurrences in the same environment.

In sum, interacting with direct nature, such as forests, parks, and mountains, produces a relaxation primarily through the kind of activities that are common in nature and that are detached from everyday life and social feedback. These positive experiences should in themselves result in restoration both when conditioned and not. However, as the article illustrates shortly, conditioning this restoration with the environment should provide an even stronger restoration in nature when both doing relaxing activities and not doing relaxing activities, as well as providing restoration for indirect nature. We also cover likely innate dispositions for several of these restorative conditionings in Section 2.2. Regularly experiencing the same core affect in an environment should lead to a conditioning of the two.

### 2.2. Restorative Conditioning

CRT suggests that conditioning restoration and natural environments to create “restorative conditioning” could be achieved through the process of evaluative conditioning. Evaluative conditioning has many similarities with classical conditioning, although it focuses more on the conditioning of emotion, mostly valence, and not behavior [46]. The classic experiment in evaluative conditioning presents a high valence face with a neutral valence face. After several pairings, the valence of the neutral face increases. Similar effects are achieved regardless of medium. This knowledge is often employed in clinical settings, where most cancer patients will develop a dislike for the food they eat during treatment, as they associate the food with poor health due to radiation and/or chemotherapy [47]. Patients are therefore served food with uncommon flavors, to minimize the impact of their food aversions. Evaluative conditioning is less affected by statistical contingency [48], is resistant to extinction, does not appear to be susceptible to modulation, and unaffected by contingency awareness [46]. Importantly, evaluative conditioning is subject to counterconditioning [46], which in a CRT framework suggests natural environments can be made non-restorative with enough stressful experiences in nature. Although valence is the main focus of evaluative conditioning, other emotions have been shown to follow the same mechanism, although this has received limited research attention (J. De Houwer, personal communication, 14 January 2016). Importantly, the other “half” of core affect, arousal, has been shown to be conditioned in the same manner as valence [49]. In summary, there is strong support for the process of conditioning of core affect with stimuli, as both valence and arousal have been conditioned. This may be applied to the context of leisure activities in nature: When participating in relaxing activities in nature, as discussed in the previous section, this relaxation can be conditioned to nature. The person does not need to be aware of the connection, as contingency awareness is not necessary for evaluative conditioning [46]. Modulation is no issue; conditioning occurs even if the two stimuli are unrelated, for example trees and lack of social feedback. Even if a person is exposed to a multitude of neutral experiences in nature, this does not affect the conditioning, as evaluative conditioning is extremely resistant to extinction [46]. This suggests that, if a person has conditioned restoration with a natural setting as a child, this conditioning will remain positive as long as multiple negative experiences are not experienced in that same environment.

Importantly, evaluative conditioning has shown that some “pairs” of conditioning are faster to condition than others. For example, conditioning poor health and food gives an immediate very strong aversion towards that specific food. Similar preexisting conditions also exist in nature, such as faster conditioning of fear to snakes than to guns [50], even if the former is realistically less dangerous. The same is most likely true for many restorative settings. For example, prospect–refuge theory [51] suggests humans have an innate liking for places where we both can have an overview of the area and still be reasonably hidden. Similar innate preexisting connections could exist, for example a faster conditioning of relaxation and forests than condition relaxation and bedrooms. This is indirectly supported by research showing that visual perception evolved to perceive natural settings [52], humans can identify scenes before any objects in the scenes are identified [53], the auditory system is intimately connected to brain regions associated with core affect [54], as well as having remarkable statistical properties that were and are likely the bedrock for our perceptual system [55]. All this indicates that parallel processes actively tweaking the proposed conditioning could be present.

In a CRT framework, these pre-existing conditions can be seen as further contributing reinforcing of the conditioning process. If a person experiences more relaxation because of innate “hardcoded” factors, this should, in a classical conditioning framework, be just another stimulus. These innate hardcoded conditions should also be able to be reinforced, or even completely overwritten, as suggested by, for example, that pet snake owners watching and touching their pets experience relaxation [56,57]. If a prehistoric family travels through a field of flowers, and the youngest gets bitten by a poisonous snake and dies, it would naturally be very detrimental to not be able to overwrite that this field of flowers is safe and restorative.

### 2.3. Conditioned Restoration

We propose conditioning as the primary mechanism in the third stage, retrieval of conditioned relaxation in the absence of the unconditioned stimulus. The “classical condition model” and “conscious expectancy” are well documented cognitive mechanisms in other parts of psychology [58]. In the classical conditioning model, the conditioning of stimuli and responses triggers the response when exposed to the stimuli. The most straightforward example is when a patient is administered analgesics to alleviate pain. After multiple treatments, the patient associates pain alleviation with the analgesic, which in turn increases the effectiveness of the drug [59]. In a nature–restoration setting, a person experiences an environment, such as a forest, and has a relaxing experience in that forest on multiple occasions. A conditioning is formed, and the next time the person visits a forest, he could experience relaxation simply by entering it. Although the classical conditioning model is primarily employed in pain management treatment, there is little reason to believe the classical condition model should only be employed as a model in pain management research. Classical conditioning has been employed and is currently employed in all types of domains and has proved effective in both humans and animals across virtually every medium.

Conscious expectancy relies on conscious anticipation of something occurring. The person expects an automatic reaction to a particular situational cue, which in turn activates the anticipated response [58]. In a CRT framework, a stressed person could enter nature in the expectation that it will be relaxing and should perceive nature as (more) relaxing as a result of this expectancy. It should be noted that this mechanism requires the subject to be aware of the process [60]. If the subject is not consciously aware of what to expect or when to expect it, expectancy cannot occur. This is contrary to the classical condition model, which does not require consciousness [60].

When people are exposed to nature such as parks or forests, conditioning should create a feeling of relaxation because this feeling has been previously conditioned through evaluative conditioning, while expectancy should create relaxation because most people expect relaxation while in nature. This is corroborated by research showing a link between place attachment and perceived environmental restorativeness: people tended to assign greater restorative qualities to places with which they had stronger emotional connections than to places with which they had weaker emotional ties [61,62,63]. It should be noted that other explanations than CRT can explain this relationship. It could be because of a stronger relationship with the environment, although teasing apart the two is difficult.

### 2.4. Stimulus Generalization

Finally, we suggest retrieval of the conditioned relaxation can also occur when exposed to associated cues. Emotional activation caused by stimuli vaguely related to the original stimulus has been found in cases of post-traumatic stress disorder [64]. Among veterans, loud noises and bangs can cause powerful stressful flashbacks [65]. Similarly, viewing images of our loved ones induces very different emotions compared to viewing images of our boss at work. Indirect stimuli replicate emotions associated with the original stimulus. This effect is believed to rely largely on classical conditioning, not expectancy [58]. In a CRT framework, core affect that has been conditioned with trees will be retrieved when exposed to a small plant. A feeling of relaxation that has been conditioned with a park could be retrieved when exposed to a window view of a park.

In summary, CRT offers a framework based on conditioning research for the restorative effects of nature. The stages of CRT are reasonably established fields in other branches of psychology, primarily within evaluative conditioning, the classical condition model, conscious expectancy, and classical conditioning. In modern society, nature is used as a leisure environment; the relaxing experiences of and in nature are conditioned with nature itself, retrieving the same feelings when exposed to nature and to stimuli that represent nature.

### 2.5. Cognitive Benefits

In a CRT framework, and similar to SRT [8], the cognitive benefits of being exposed to nature rely on the elicited reduction of psycho-physiological stress, which, as we view stress in a core affect framework, implies an increase in feelings of relaxation. Consequently, CRT views “restoration” as the positive affective responses associated with stress reduction elicited by nature exposure. Because restoration is often operationalized through various psychometrics, we briefly cover how stress has been shown to influence these same psychometrics. High-anxiety subjects scored significantly lower on a digit span test than low-anxiety subjects [66]. A significant increase in error rates and response times has been shown on the socially evaluated cold pressor test, especially for tasks dependent on top-down processing following a stressor [67]. Support for ART [5] argues that nature only affects the executive control subscore of the attention network test (ANT) because, from an ART perspective, nature only affects attention. We argue that this argument is flawed because to our knowledge no experimental study has ever been able to affect the two remaining subscores, alerting and orienting. Short-term traditional Chinese meditation training resulting in lower anxiety, and stress-related cortisol levels, only showed significant effects on executive control in the ANT [24]. Heating participants in a 50 °C room for 60 min also only affected executive control [68], even though heat stress exposure has been shown to cause stress [69]. Thus, even though heating and meditation affect stress, they did not affect alerting and orientation subscores of the ANT. This could imply that alerting and orienting are extremely stable and very hard to affect, and that stress affects only the executive control subscore of the ANT.

There is an abundance of studies or theories regarding stress and cognitive performance. We suggest that stress and stress reduction is likely the core mechanism behind the differences in test scores observed in most restorative nature studies, as studies show a negative correlation between stress and test scores. It is this mechanism CRT also relies on to explain the same findings.

The effect of nature on emotional psychometrics far outweighs its effect on attentional measurements, suggesting that emotions could mediate the relation between nature and attentional scores. Meta-analysis has shown support for exposure to nature’s effect on emotions but only some support for greater attention after exposure to a natural environment, though not after adjusting effect size for pretest differences [26]. Another meta-analysis showed that test scores measuring working memory, cognitive flexibility, and, to a less reliable degree, attentional control may be improved after exposure to nature [70], but, when comparing this effect size to effect sizes on affective measurements, we see that effect sizes on the latter are larger. Brief exposure to natural environments had an effect size of Cohen’s D = 0.652 on positive affect, and D = 0.24 on negative affect [71], while the overall effect size on attention had been shown to be Hedges G = 0.162 [70]. While the two effect sizes do not represent the exact same metric, they are very comparable and suggest that nature’s effect on affect is stronger than its effect on attention. This should indicate that nature’s effect on emotions is stronger than its effect on tests of attention and other cognitive functions, which could, in turn, in a CRT framework, indicate that nature’s effect on various test scores is primarily mediated by its effect on core affect.

In a CRT framework, nature can also be restorative because of the absence of other environments. Just as relaxation can be conditioned with a forest, stress can be conditioned with a subway. For example, on a subway one can repeatedly experience being in a rush, which causes stress. The negative emotions produced by this will condition with the subway itself and be re-experienced when exposed to subways. The very act of not being in a stressful environment will in itself “lower”, as in not increase, stress levels. Additionally, conditioning is not the only environmental aspect that can influence core affect. For example, the role of appraisal has also been shown to influence core affect, where appraisal of whether something is advantageous or disadvantageous, caused by others, has a desired expected outcome, or whether an emotional coping style can be applied has been shown to influence core affect [72]. As appraisal affects core affect, which includes relaxation, which again CRT argues is restoration. It follows that a CRT framework allows for other mechanisms other than conditioning to trigger restoration. For example, appraising that a situation has a desired expected outcome, should contribute to restoration.

### 2.6. Empirical Support

We can find support for a CRT framework in different existing studies. CRT postulates that any environment could theoretically produce the same effect as nature (although difficult due to innate dispositions), and that there are variations between individuals based on previous interactions with the environment. Participants staying in a hospital isolation room redecorated in a traditional homey fashion felt thermally more comfortable and, more importantly, plasma cortisol levels—a common indicator of stress—decreased significantly compared to control conditions, especially for participants with high pretest levels [18]. Participants with lower stress levels showed no improvement, implying that the stress level tends *towards* a specific level, likely the conditioned low-stress level of home. Students who walked in nature had significantly lower plasma cortisol levels compared with students who walked in a gym or watched a nature video during exam periods, but not during no-exam periods [73]. Again, this finding suggests that stress levels tends towards a pre-conditioned level.

Museum visitors have also been shown to receive some level of cognitive restoration [17]. Museums represent an environment that strongly encourages relaxing activities and are perhaps closer to nature in terms of experience than most other non-natural environments. The study found that museum “novices” did not experience much restoration. CRT suggests that they do not receive the same amount of restoration simply because the museum setting has not been conditioned in these participants. It is important to note that other explanations than CRT could explain the same finding. It could be that people who do not enjoy museums never return, so that museum “veterans” have self-selection towards enjoying the museum and therefore experience restoration. The new visitors could also consist of a different demographic, or be less enthusiastic about museums, which again result in lower levels of restoration. Additionally, a composite score of plant, bird, and bee/butterfly species richness, habitat number in parks have been shown to correlate with perceived restorative benefit [74]. In a CRT perspective, these biodiversity-related variables should be highly correlated with the number of stimuli providing conditioned relaxation, such as the humming of bees, chirping of birds, and sightings of various plants.

CRT claims that any associated stimuli can trigger the relaxation conditioned to an environment. After performing a stressful mental arithmetic task, subjects were exposed to nature sounds, high noise, low noise, or ambient noise. The nature sounds reduced their skin conductance levels, which is interpreted as sympathetic nervous system activation, which in turn is interpreted as stress [22]. These data could be said to be particularly problematic to a visually focused theory, as restoration occurs in the absence of visual stimuli. These findings corroborate CRT’s assumption that conditioned restoration can occur with any sensory input.

ART, SRT, and fractal theories postulate that restorative processes occur via visual contact with nature, viewing nature in VR should induce equal or very similar effects. Starting with Ulrich’s classic study [11], evidence exists showing that viewing nature in pictures or videos can elicit stress-reducing or restorative effects. Studies have shown that viewing nature scenes in immersive virtual reality (VR) is more effective than viewing nature on a screen in eliciting a faster reduction of anxiety/stress and increased positive affect [75]. However, to date, the evidence on the restorative benefits of VR-based nature exposure is mixed and, in general, indicative of greater effects induced by exposure to real environments as compared with virtual environments [76]. Interesting, some studies have found that, as opposed to a real natural environment, exposure to the same natural environment through VR may not induce changes in positive or negative affect, in spite of the fact that the real and virtual environment was rated with equivalent levels of perceived restorativeness potential [77]. Similar findings were confirmed also when adjustments to the virtual environments were done to reduce confounders such as cyber-sickness [78]. On the one hand, the reduced restorative power of virtual nature, as compared to real nature, is likely to be, at least in part, explained by an incomplete likeness—e.g., immersive virtual environments have limited image-sharpness and, especially when consisting in static images, lack of movements such as waving of leaves or moving of clouds in the sky. On the other hand, these differences betray the limit of understanding nature-induced restorativeness as exclusively mediated by visual cues: smells, sounds, the fresh air, or warm sun on the skin, and many other elements are likely to play an important role in the restorative process. Moreover, the smaller magnitude of the restorative effects induced by exposure to virtual nature is more in line with a CRT framework, where cues of the conditioned stimulus recall emotional responses similar to those experienced in real nature.

According to ART, SRT and fractal theories, nature possess visual characteristics that should elicit restoration in any individuals irrespectively of how “used” they are to be in contact with nature or what are their conscious beliefs and values relative to nature. Restorative effects are found to a lesser extent in people who are repeatedly in contact with nature for non-leisure purposes. Children who had a work relationship with nature reported less restoration when spending free time in nature than children who do not work in nature [79]. Similar results are shown in reported restoration and occupational engagement [80], Attention-restoration effects were not found among elderly living in nature-rich Ireland’s country-side [81,82]. On the other hand, studies have shown that changes in positive affect induced by nature exposure are mediated by the individuals’ feelings of connectedness with nature [83,84]. More in general, the role of nature connectedness in the complex human–nature interactions, has been emerging as an important perspective that can not only influence the extent to which people view nature, but also may by itself explain individuals psychological wellbeing [85]. The lack or reduced restorative effects in association with nature exposure among individuals who are often in contact with nature in non-leisure contexts, is in contradiction with ART, SRT and fractal theories, which assign to nature intrinsic characteristics that are expected to induce restoration irrespectively people’s previous experiences with nature. This evidence is rather in line with a CRT framework, which postulates that the restorative effects of nature are the result of a conditioned response elicited by previous pairing with leisure experiences. On the other hand, even though CRT still is focused on what is probably only a small piece of the complexity of human–nature interactions, it can to a certain extent account for people’s copious beliefs and values they assign to the natural world (e.g., feelings of nature connectedness). This may, for example, occur in form of anticipated psychological benefits (or even placebo effects). Additionally, CRT claims that preferences for different landscapes should exist between different populations exposed to different landscapes in their leisure time. When investigating preference ratings on pictures of nature with different levels of fractal values, preference for different fractal values differs amongst countries and levels of urbanness [86]. While this is hard to explain in any purely innate framework, these findings make perfect sense in a CRT framework. Different populations have been conditioned to prefer different landscapes and thus different levels of fractal values.

CRT also claims that there should be several connections to nature experiences in child and adulthood. Frequency of childhood visits to nature has been shown to be correlated to both the frequency of visits as an adult, agreeing more with positive statements about green places such as making them feel “more energetic” and being “magical” [87], and the likelihood of engaging in higher levels of natural environment-based physical activity in adult life [83]. Participation with wild and domesticated nature in childhood has also shown to correlate with environmental attitudes and behaviors in adulthood [88]. While these studies cannot prove a causal relationship because they are non-experimental retrospective, they all suggest a likely causal connection between experience with nature in child and adulthood. Designing studies that provide evidence for these interrelationships are challenging.

### 2.7. Integrating CRT with Established Theory

CRT aims to account for some of the cognitive, affective, and physiological mechanisms explaining how nature provides restoration. The theory is proposed as an additional framework that can contribute to explaining the complex relation between humans (and sentient living beings in general) and the environment, possibly in integration with and/or addressing some of the limitations of existing theories. CRT is in line with SRT’s [8] suggestion that mechanisms that adjust stress levels are behind the primary benefit of nature. However, CRT does not agree with the pure direct unmediated evolutionary explanation that nature has stress-reducing qualities in itself. To the opposite, CRT proposes that stress reduction primarily happens when relaxation has been previously conditioned with nature, so that later exposure to nature retrieves the conditioned feelings of relaxation, resulting in stress reduction (see Figure 1). CRT suggests that this stress reduction mechanism is the primary cause of the physical benefits from exposure to nature, through the activation of the parasympathetic nervous system activity which promotes “rest-and-digest”, as discussed in the Introduction.

While acknowledging that natural environments often possess intrinsic characteristics that make them particularly advantageous for eliciting psychophysiological restoration (e.g., by providing quietness and opportunities for getting away from daily hassles), in contrast with ART, CRT does not see attentional restoration as a direct effect of nature, but suggests that attention restoration is rather an effect of stress reduction, as the psychometric used to demonstrate attention restoration show the same change in stress-related studies, as discussed in Section 2.5. CRT does, however, acknowledges that the environmental qualities described in ART (i.e., being away, compatibility, fascination, and extent) correlate strongly with restoration, and could be an ideal psychometric concerning the phenomenon, because these factors strongly correlate with how core affects are associated with nature exposure. Being away and fascination, in particular, should predict a high-valence low-arousal connection which facilitates restoration according to both ART and CRT. On the other hand, emotional responses to environments to which positive emotional states are conditioned can lead to greater perceptions of coherence and compatibility [61], contributing to stronger restorative experiences. These environmental qualities might indeed contribute to create the conditioned restoration stimuli. For example, individuals are more likely to perform leisure activities in environments that match their preferences and inclinations at a given time (thus environments with high levels of “compatibility”). The feelings of “being away” associated with leisure activities, can facilitate the primary (unconditioned) stress reduction experience, which will be subsequently conditioned to the environment. Similarly, the environmental qualities “fascination” and “extent” (which often characterize natural environments) will also be conditioned to the restorative response. It is important to note that we do not suggest the theories as exclusive. It is fully possible for a mechanism to apply to nature, but not apply to other environments. Different processes can result in the same outcome. Just as ART is not “competing” with SRT [23], CRT offers primarily to explore a previously unexplained mechanism behind restoration, not discredit all other explanations.

While the most important aspect in relation to CRT is to argue that leisure *is* relaxing, and thus possibly restorative, it is also useful to examine why leisure is relaxing. According to the polyvagal theory [13], two existing vagal primitive neural circuits promote defensive strategies, and one promotes growth and restoration. The latter circuit promotes social behavior, communication, and visceral homeostasis, and are incompatible with activation of neurophysiological systems responding to defensive situations. To switch system, the individual must first assess risk, a process polyvagal theory suggests does not require consciousness, and thus dubs it “neuroception” [13]. Neuroception of familiar individuals, appropriately prosodic voices and warm expressive faces promote this safety, which again activates the neural circuit promoting growth and restoration. We suggest that familiar individuals, appropriately prosodic voices and warm expressive faces are closely associated with leisure activity, and will especially be the case in nature or other low-population areas where there is an absence of “non-warm faces”, such as people in a rush. This could be the neurological explanation for how leisure and relaxation create restoration.

Conditional mechanisms have also been subject to neurological research, and it is these same mechanisms CRT relies on. Fear conditioning, perhaps the best researched form of conditioning, has shown a robust pattern of neural activation in humans [89] and clear evidence of the central involvement of the amygdala in both animals and humans [90]. The specifics of evaluative conditioning have also received attention, but a consensus in the field is not yet reached. For example, unilateral amygdaloid nuclear complex impairment has shown to both inhibit and not inhibit the mechanism [91,92]. When the neurological basis for evaluative conditioning is further established, CRT should follow.

### 2.8. CRT Within the Broader Context of a Two-Sided and Sustainable Human–Nature Relationship

In recent years, researcher have been acknowledging that focusing on the restorative effects of nature is, at best, too simplistic [93]. In this perspective, the human–nature relation can easily be seen as a one-sided trade, where natural environments are investigated in the extent to which they can benefit people, often with the purpose of understanding how to best design ad-hoc environments that can maximize human health. While this may address important issues, such as how to promote health and well-being in the population, the risk is to see and use nature as a commodity. The benefits of the human–nature relationship go well beyond the acute restorative effects induced by nature exposure: besides the fact that a balanced equilibrium of the natural ecosystem is essential for life on this planet, people’s conscious feelings of connectedness with the natural world (e.g., the extent to which people feel to be part of the natural world, as opposed to feeling separated by it) has been shown to be associated with health and mental well-being [85]. Hence, the inherent risk in focusing solely on the mechanisms that lead to restoration, is that this may lead to deterioration of the natural world (e.g., loss of biodiversity in urban green spaces, in favor of more aesthetically pleasing and “people-friendly” nature elements) as well as deterioration of authentic feelings of nature connection (e.g., progressively unlearning to appreciate the richness of biodiversity). Researchers have thus been calling for interdisciplinary approaches and theoretical frameworks that can more comprehensively account for the complexity of a two-sided and sustainable human–nature relationship [15,93,94].

CRT could allow for systematically enhancing emotional connectedness to nature. In the present review, CRT has been so far presented mainly as a theoretical framework to explain the restorative effects of nature exposure. Therefore, not unlike ART, SRT, and perceptual fluency, this perspective still presents limitations in the extent to which it can encompass the complexity of a two-sided and sustainable human–nature relationship as above described. We argue, however, that CRT may have, to some extent, the potential of serving as a more flexible theoretical framework and useful supplementary tool in this respect. For example, CRT-driven programs may be developed to pair enjoyable experiences with exposure to environments rich in biodiversity, with the purpose not only to promote restorative experiences, but also (and foremost) to enhance emotional connectedness and appreciation of the natural environment in its full complexity. Some evidence exists suggesting that such approaches may be effective. As little as a 10-min walk in an arboretum was found to lead to enhanced feelings of nature connectedness [84]. Pupils who had more opportunities for nature contact within school-context were found to be more empathetic and concerned for non-human life forms, as well as more cognitively aware of human–nature interdependence [95]. Biodiversity-focused outdoor learning programs were found to elicit psychological benefits as well as enhanced feelings of nature connectedness among school children [96]. CRT may serve also as a framework to understand how to de-condition negative emotions associated with certain natural environments or nature elements. Studies have, for example, proven the effectiveness of controlled exposure to reduce fear and of brown bears and increased their social trust in wildlife management authorities [97].

### 2.9. Limitations

CRT’s primary weakness is that most of the evidence it relies on does not directly relate to nature–restoration settings. No thorough investigations have been conducted of evaluative conditioning, the classical condition model or conscious expectancy in a relaxation–nature pairing. Evaluative conditioning could be especially criticized for relying heavily on lab experiments with limited real-life findings [98]. Many studies are needed to show that the mechanisms CRT suggests also apply to nature–relaxation, and not just in other fields.

CRT suggests that the primary reason nature is and become restorative, is because it is more detached from everyday life than almost any other environment. This detachment allows for conditioning of positive stimuli with an absence of stressful stimuli in nature unrivaled to other environments. This also allows the framework to be adapted to other environments, such as offices inducing stress if your office job makes you stressed. A slight weakness of the model is that innate preferences for nature have indeed been suggested in studies, possibly changing the conditioning mechanics CRT relies on. Conditioning a fear of snakes is easier than conditioning a fear of guns [50] and even activates evolutionary brain networks [99], as well as a cross-cultural preference for trees with a wide canopy width exists [100], suggesting innate conditions exist. Other theories also suggest innate mechanisms for restoration in nature exist, but focus on soundscape, arguing that the auditory system is not well matched to our current habitat, and thus fails in communicating safety [54]. This could also be a valid reason nature is more restorative than non-natural environments, but we argue that the two can also easily coexist, that nature is restorative both through soundscape appraisal and conditional mechanisms. Because the mechanisms CRT suggests have not been tested in a nature–relaxation pairing, we cannot be certain on how these likely innate dispositions affect the proposed conditioning.

## 3. Conclusions and Future Research

The present paper discusses how CRT can contribute to explaining the mechanisms underlying environmental restorativeness. Empirical evidence confirming the formation of conditioned restorative stimuli is needed to provide support for this theory. In particular, we recommend longitudinal studies where a restorative experience is conditioned with non-natural environments, such as monasteries or museums, and that restoration in these environments, as well as associated stimuli, be examined over time and compared to a control group with no conditioning. Further research could also investigate whether experience with nature in childhood affects the emotions nature creates later in life. Finally, it would also be interesting to investigate whether people with work relationships with specific natural environments, such as parks, experience restoration in other natural settings, such as lakes, compared with control groups with no work relationship with any natural setting.

In summary, CRT offers a framework based on conditioning that can be applied to investigate the restorative effects of nature. Most of the individual stages of CRT are established fields in other branches of psychology, primarily within evaluative conditioning, placebo research, core affect, and classical conditioning. In modern society, nature is used as a leisure environment. The relaxing experiences of and in nature are conditioned with nature itself, which gives emergence to the same relaxation when exposed to nature, as well as items that represent nature. We believe the theory will contribute to further insight into the mechanisms driving the psychological and physiological recovery exerted by natural environments, as well as other environments. This could improve design of areas where recovery is of great importance, such as the healthcare sector.

## Figures and Tables

**Figure 1 ijerph-17-06792-f001:**
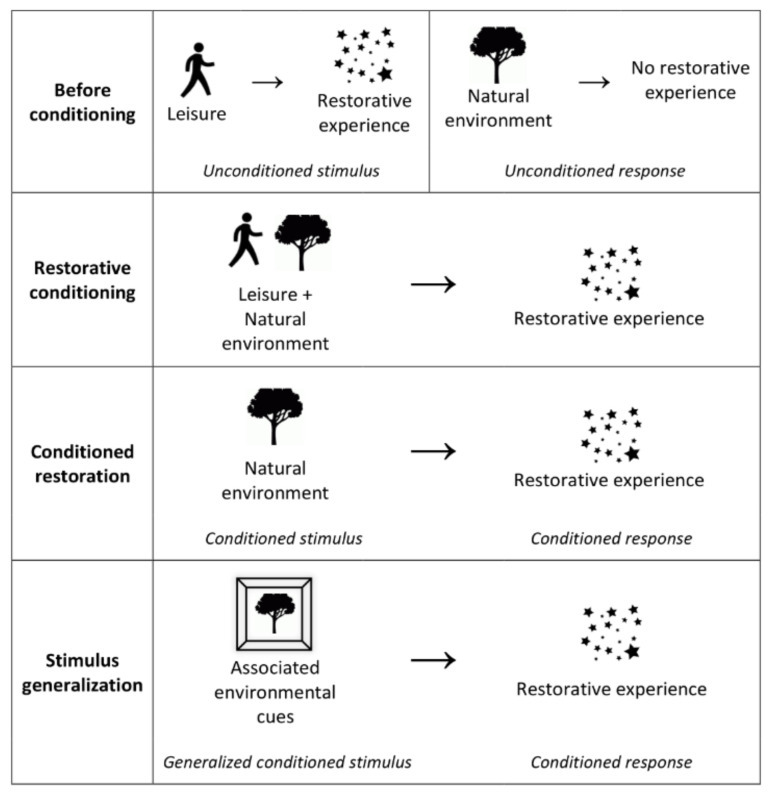
A schematic model of conditioned restorative theory (CRT) in a leisure–nature–restoration scenario.

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
