# Peer review of "Proposing a Framework for the Restorative Effects of Nature through Conditioning: Conditioned Restoration Theory"

_ijerph, 2020, doi:10.3390/ijerph17186792_

Round 1

Reviewer 1 Report

The manuscript addresses the topic of great interest: the psychological and physiological recovery exerted by natural environments.

The introduction offers a fairly complete picture on the issue of research and directly related topics, but the literature analysis is not particularly up-to-date. In particular, using the keywords indicated by the authors, it is easy to identify more recent studies (as of 2016) that have not been covered.

From the methodological viewpoint, the paper does not provide any indication of the method followed that can be guessed in the course of reading. It would be desirable for the authors to dedicate at least a few lines to describe what way to introduce the new theoretical approach to the issue. Is it a theoretical building?

The remaining parts of the paper are very clear and the authors' positions well argued, although the conclusions are perhaps a little too concise with respect to the relevance that the proposed new theoretical approach might have.

Finally, in order not to leave any doubt about the originality of this paper, it would be appropriate for the authors to clarify how their research compares to another manuscript that can be easily found on the web with a very similar purpose:

Exploring the restorative effects ofenvironments through conditioning - The conditioned restoration theory

https://brage.inn.no/inn-xmlui/bitstream/handle/11250/2426176/MPSY3010%20Masteroppgave%20Lars%20Even%20Egner%20september%202016.pdf?sequence=1

Since it is only a matter of perfecting the manuscript, I think I can attribute a minor revision before its publication.

Reviewer 2 Report

I don't understand the idea of the paper. Is it a literature review, or research paper? Furthermore, authors declared that it theoretically not unique to nature. So what is the novelty of this paper? If it is a literature review, authors should rewrite the paper and suggest the new idea for future researchers.

Reviewer 3 Report

The purpose of this paper is to
present the Conditioned restoration theory (CRT) as a theoretical framework that may contribute to explaining the mechanisms underlying the restorative effects of nature, as well as other
environments. This paper describe CRT, discuss its strengths and limitations, review research literature that supports it, and proposed how it can be integrated with established theories.

Minor revisions: It is a very interesting paper and for health x environment/culture. I would like to know if it is more papers that could be cited, more examples. The paper is well-done and I suggest the publication after minor revisions.

Reviewer 4 Report

I appreciate the opportunity to read this manuscript and commend the authors for their knowledge and clear explanation of the dominant theories associated with nature restoration and research as well as their efforts to contribute to this area of research.  In general, the write up seems thorough and well-referenced, combining foundational literature and current literature.

The subject matter is timely as people learn more and debate the place of the natural environment for the wellbeing of humans. Additionally, mental health issues are a global concern. Please use my remarks to strengthen some of the arguments in your paper.

The authors did well bringing in critiques on two long-standing current theories on restoration.  It is useful to question the psycho-evolutionary perspective of ART and SRT and thus far there is no way to prove these may have evolved as adaptive traits.  However, one does not want to throw the baby out with the bathwater. It is as hard to prove these have not evolved as adaptive traits.  Additionally, the authors pointed out that perhaps these theories are related, not mutually exclusive and may work together, indicating a potential problem in academia of trying to split hairs and argue for a certain theory at the risk of not seeing the forest for the trees.

Because ART does not, as the authors said, “explain some of the broader psycho-physiological effects of nature exposure, such as quicker hospital recovery time [10], reduced blood pressure, anger and aggressiveness, reduced arousal measured by cardiac interbeat interval [21], and lower sympathetic activation measured through skin conductance level” does not necessarily support a fallacy in ART, but perhaps there is more than ART going on in regards to human’s relationship with nature. Again perhaps ART and SRT work together.

A strong takeaway is that there is more to learn about these specific theories as well as human and nature relationships. Their critique of ART and SRT open the door that there could be alternative explanations for the effects seen on human psychology and physiology in regards to human and nature relationships.

In that same vein, proving the validity of CRT does not show that ART and SRT are inaccurate, just that CRT may work. Associating nature or other things with relaxation and therefore triggering relaxations makes sense for some people.  However, I do not agree that the vast majority of members of modern society interact with nature in relaxing leisure.  People not taking the time to interact with nature (the vast majority of people) is often stated a is a cause for concern. Numerous people interact with nature when they have to during disasters such as fires, hurricanes, tornados, floods, and the like.  Many people interact with nature in leisure activities but that is not likely the majority of the population.

Another weakness in the author’s argument for CRT is that I am not sure it is logical that if a person works in a natural environment they cannot or are not as likely to be restored by it.  This does not seem to be true for outdoor guides, for example. More likely the activity the guide teaches may not be as restorative as it used to be, however, many people purposefully engage in work in the outdoors because they enjoy and feel better when working in the outdoors.

Other authors have agreed that repeated time in nature when a child often leads to those people spending time in nature as an adult—certainly that could be classical conditioning model—which may be a part of the phenomenon, though it may not be the primary cause or it could be primary.  Sorting out the interrelationships are challenging.

Humans are wired for forming relationships and need to be in relationships for psychological and physiological health. When people are exposed to nature during outdoor recreation events some people argue that something deep in people’s consciousness is awakened—a time when humans lived in nature relating on a daily basis.  It is hypothesized that humans return to outdoor recreation because of that consciousness that is kindled or rekindled. Additionally, many Indigenous people practice relationships with the natural world as Robin Kimmerer has written about. It is about developing a healthy relationship, which, again does not discount that ART, SRT, and CRT may also exist, though Kimmerer and others suggest that looking at a slice when there is a system of relationships at work, may not be fruitful.

From the authors: “people tend to assign greater restorative qualities to places with which they have stronger emotional connections than to places with which they have weaker emotional ties [57–59].” This may be about the strength of a relationship, rather than conditioning—though it would be hard to tease these apart.

In this manuscript, the grammar is generally sound; there are some typos. There are some writing problems such as a possible spelling error:  artefactual. Line 472-3: ART (i.e,  being  away, compatibility, fascination, and extent): there needs to be a period after the e.  Names of theories are not capitalized.  Line 148 has an extra period, the page number goes after the quote, and literature is reported in the past tense.

As the authors make transitions they often use the word WILL.  Such as line 237: But as the article will illustrate, conditioning… Line 240: We will also cover likely …  Because the paper is written the present tense should be used.  Please change throughout the paper.

Thank you, again for the opportunity to review this manuscript. It should help inspire discussion about human and nature relationships.

Round 2

Reviewer 2 Report

Accept